# Diffusion Model Predictive Control

**Guangyao Zhou***  *stannis@google.com*
*Google DeepMind*

**Sivaramakrishnan Swaminathan**  *sivark@google.com*
*Google DeepMind*

**Rajkumar Vasudeva Raju**  *rajvraju@google.com*
*Google DeepMind*

**J. Swaroop Guntupalli**  *swaroopgj@google.com*
*Google DeepMind*

**Wolfgang Lehrach**  *wpl@google.com*
*Google DeepMind*

**Joseph Ortiz**  *joeortiz@google.com*
*Google DeepMind*

**Antoine Dedieu**  *adedieu@google.com*
*Google DeepMind*

**Miguel Lázaro-Gredilla**  *lazarogredilla@google.com*
*Google DeepMind*

**Kevin Murphy**  *kpmurphy@google.com*
*Google DeepMind*

**Reviewed on OpenReview:** *https://openreview.net/forum?id=pvtgffHtJm*

## Abstract

We propose Diffusion Model Predictive Control (D-MPC), a novel MPC approach that learns a multi-step action proposal and a multi-step dynamics model, both using diffusion models, and combines them for use in online MPC. On the popular D4RL benchmark, we show performance that is significantly better than existing model-based offline planning methods using MPC (e.g. MBOP(Argenson & Dulac-Arnold, 2021)) and competitive with state-of-the-art (SOTA) model-based and model-free reinforcement learning methods. We additionally illustrate D-MPC's ability to optimize novel reward functions at run time and adapt to novel dynamics, and highlight its advantages compared to existing diffusion-based planning baselines.

## 1 Introduction

Model predictive control (MPC), also called receding horizon control, uses a dynamics model and an action selection mechanism (planner) to construct "agents" that can solve a wide variety of tasks by means of maximizing a known objective function (see e.g., Schwenzer et al. (2021) for a review of MPC). More precisely, we want to design an agent that maximizes an objective function $J(a_{t:t+F-1}, s_{t+1:t+F})$ over a planning horizon

---

*Corresponding author: stannis@google.com

$F$ from the current timestep $t$

$$\boldsymbol{a}_{t:t+F-1} = \arg\max_{\boldsymbol{a}_{t:t+F-1}} \mathbb{E}_{p_d(\boldsymbol{s}_{t+1:t+F}|\boldsymbol{s}_t, \boldsymbol{a}_{t:t+F-1}, \boldsymbol{h}_t)} \Big[ J(a_{t:t+F-1}, s_{t+1:t+F}) \Big], \tag{1}$$

where $\boldsymbol{h}_t \equiv \{\boldsymbol{s}_{1:t-1}, \boldsymbol{a}_{1:t-1}\}$ is the history. MPC thus factorizes the problem that the agent needs to solve into two pieces: a modeling problem (representing the dynamics model $p_d(\boldsymbol{s}_{t+1:t+F}|\boldsymbol{s}_{1:t}, \boldsymbol{a}_{1:t+F-1})$, which in this paper we learn from offline trajectory data) and a planning problem (finding the best sequence of actions for a given reward function). Once we have chosen the action sequence, we can execute the first action $\boldsymbol{a}_t$ (or chunk of actions) and then replan, after observing the resulting next state, thus creating a closed loop policy.

The advantage of this MPC approach compared to standard policy learning methods is that we can easily adapt to novel reward functions at test time, simply by searching for state-action trajectories with high reward. This makes the approach more flexible than policy learning methods, which are designed to optimize a fixed reward function.[1] In addition, learning a dynamics model is often more sample-efficient than learning a policy directly (Zhu et al.). This is because dynamics model training is essentially a supervised regression problem, predicting the next state given the current state and action—a mapping that is typically well-behaved and near-deterministic. Policy learning, however, involves predicting actions, where the optimal behavior may be multimodal (multiple good actions exist) and require accurate long-horizon credit assignment, making it a more complex learning task given the same data budget. Finally, dynamics models can often be adapted more easily than policies to novel environments, as we show in our experiments.

However, to make MPC effective in practice, we have to tackle two main problems. First, the dynamics model needs to be accurate to avoid the problem of compounding errors, where errors in next state prediction accumulate over time as the trajectory is rolled out (Venkatraman et al., 2015; Asadi et al., 2019; Xiao et al., 2019; Lambert et al., 2022). To avoid compounding errors, multi-step models are preferable. However, these require a model class capable of capturing the complex, multimodal distribution of entire trajectories. This motivates our use of diffusion models. Second, the planning algorithm needs to be powerful enough to select a good sequence of actions, avoiding the need to exhaustively search through a large space of possible actions.

We tackle both problems by using diffusion models to learn joint trajectory-level representations of (1) the world dynamics, $p_d(\boldsymbol{s}_{t+1:t+F}|\boldsymbol{s}_t, \boldsymbol{a}_{t:t+F-1}, \boldsymbol{h}_t)$, which we learn using offline "play" data (Cui et al., 2023); and (2), an action sequence proposal distribution, $\rho(\boldsymbol{a}_{t:t+F-1}|\boldsymbol{h}_t)$, which we can also learn offline using behavior cloning on some demonstration data. Although such a proposal distribution might suggest actions that are not optimal for solving new rewards that were not seen during training, we show how to compensate for this using a simple sampling-based planner, a variant of the random shooting method that uses a multi-step diffusion model trained on offline datasets as the action proposal and a simple alternative to more complex methods such as trajectory optimization or cross-entropy method. We call our overall approach *Diffusion Model Predictive Control* (D-MPC).

We show experimentally (on a variety of state-based continuous control tasks from the D4RL benchmarks (Fu et al., 2020)) that our proposed D-MPC framework significantly outperforms existing model-based offline planning methods, such as MBOP (Argenson & Dulac-Arnold, 2021), which learns a single-step dynamics model and a single-step action proposal, and hence suffers from the compounding error problem. By contrast, D-MPC learns more accurate trajectory-level models, which avoid this issue, and allow our model-based approach to match (and sometimes exceed) the performance of model-free offline-RL methods. We also show that our D-MPC method can optimize novel rewards at test time, and that we are able to fine-tune the dynamics model to a new environment (after a simulated motor defect in the robot) using a small amount of data. Finally we perform an ablation analysis of our method, and show that the different pieces — namely the use of stochastic multi-step dynamics, multi-step action proposals, and the sampling-based planner with learned action proposals — are each individually valuable, but produce even more benefits when combined.

In summary, our key contributions are:

---

[1] Goal-conditioned reinforcement learning (RL) can increase the flexibility of the policy, but the requested goal (or something very similar to it) needs to have been seen in the training set, so it cannot optimize completely novel reward functions that are specified at run time.

1. We introduce Diffusion Model Predictive Control (D-MPC), combining multi-step action proposals and dynamics models using diffusion models for online MPC.

2. We show D-MPC outperforms existing model-based offline planning methods on D4RL benchmarks, and is competitive with SOTA reinforcement learning approaches.

3. We demonstrate D-MPC can optimize novel reward functions at runtime with a simple sampling-based planner, and adapt to novel dynamics through fine-tuning.

4. Through ablations, we validate the benefits of our method's key components individually and in combination.

## 2 Related work

Related work can be structured hierarchically as in Table 1. Model-based methods postulate a particular dynamics model, whereas model-free methods, whether more traditional —behavioral cloning (BC), conservative Q learning (CQL) (Kumar et al., 2020), implicit Q-learning (IQL) (Kostrikov et al., 2021), etc— or diffusion based — diffusion policy (DP) (Chi et al., 2023), diffusion BC (DBC) (Pearce et al., 2023)— simply learn a policy. This can be done either by regressing directly from expert data or with some variant of Q-leaning. Model-based approaches can be further divided according to how they use the model: Dyna-style (Sutton, 1991) approaches use it to learn a policy, either online or offline, which they can use at runtime, whereas MPC-style approaches use the full model at runtime for planning, possibly with the guidance of a proposal distribution.[2]

It is possible to model the dynamics of the model and the proposal jointly using $p_j(s, a)$ or as factorized distribution $p_d(s|a)\rho(a)$. The latter allows for extra flexibility, since both pieces can be fine-tuned or even re-learned independently. Finally, we can categorize these models as either single-step (SS) or multi-step (MS). SS methods model the dynamics as $p_d(\boldsymbol{s}_{t+1}|\boldsymbol{h}_t, \boldsymbol{a}_{t+1})$ (where $\boldsymbol{h}_t = (\boldsymbol{s}_{t-H:t}, \boldsymbol{a}_{t-H:t})$ is the history of length $H$), and the proposal as $\rho(\boldsymbol{a}_t|\boldsymbol{h}_t)$, so we predict (a distribution over) the next time step, conditioned on past observations (and the next action, for the dynamics). We extend this to the whole planning horizon of length $F$ by composing it in an autoregressive form, as a product of one-step conditionals, i.e., $p_d(\boldsymbol{s}_{t+1:t+F}|\boldsymbol{s}_{1:t}, \boldsymbol{a}_{1:t+F-1}) = \prod_t^{t+F-1} p_d(\boldsymbol{s}_{t+1}|\boldsymbol{s}_t, \boldsymbol{a}_t, \boldsymbol{h}_t)$. (Note that this might result in compounding errors, even if $\boldsymbol{h}_t$ contains the entire past history, i.e., is non-Markovian.) By contrast, multi-step (MS) methods model the joint distributions at a trajectory level. Thus the MS dynamics model represents $p_d(\boldsymbol{s}_{t+1:t+F}|\boldsymbol{s}_t, \boldsymbol{h}_t, \boldsymbol{a}_{t:t+F-1})$ and the MS proposal represents $\rho(\boldsymbol{a}_{t:t+F-1}|\boldsymbol{s}_t, \boldsymbol{h}_t)$.

Several recent papers follow the SS Dyna framework. Some using traditional dynamics modeling (e.g., MOREL (Kidambi et al., 2020), MOPO (Yu et al., 2020), COMBO (Yu et al., 2021), RAMBO-RL (Rigter et al., 2022) and Dreamer (Hafner et al., 2020)), and others using diffusion. The latter includes "Diffusion for World Modeling" paper (Alonso et al., 2024) (previously called "Diffusion World Models" (Alonso et al., 2023)), "UniSim" paper (Yang et al., 2024), and the "SynthER" paper (Lu et al., 2024). These are then used to generate samples from the model at training time in order to train a policy with greater data efficiency than standard model-free reinforcement learning (RL) methods. Some other recent papers — such as "Diffusion World Model" (Ding et al., 2024), "PolyGRAD" (Rigter et al., 2024) and "Policy-Guided Diffusion" (Jackson et al., 2024) — have proposed to use diffusion for creating joint multi-step (trajectory-level) dynamics models. However, being part of the Dyna framework, they are not able to plan at run-time, like D-MPC does.

There are many papers with a model-based approach. Probably the closest to our work is "Diffuser" (Janner et al., 2022), which uses diffusion to fit a joint (state, action) multi-step model $p_j(\boldsymbol{s}_{1:T}, \boldsymbol{a}_{1:T})$ using offline trajectory data. They then use classifier-guidance to steer the sampling process to generate joint sequences that optimize a novel reward at test time. The main difference to our method is that we represent the joint as a product of two models, the dynamics $p_d(\boldsymbol{s}_{1:T}|\boldsymbol{a}_{1:T})$ and the policy / action proposal, $\rho(\boldsymbol{a}_{1:T})$. As we show in Section 4.3, this factorization allows us to easily adapt to changes in the world (e.g., due to hardware defects) from a small amount of new data, whereas Diffuser struggles in this context. In addition, we propose

---

[2]Note that a proposal distribution (which we denote by $\rho(a)$) is different than a policy (which we denote by $\pi(a)$), since rather than determining the next best action directly, it helps accelerate the search for one.

| | Factored: $p_d(s|a)\,\rho(a)$ | | | | Joint: $p_j(s,a)$ | Model-free: $\pi(a)$ |
|---|---|---|---|---|---|---|
| | Dyna | | MPC | | MPC | |
| | (single-step) | (multi-step) | (single-step) | (multi-step) | (multi-step) | |
| Examples | MOReL etc., Dreamer, DWMS, UniSim, SynthER | DWM, PolyGRAD PGD | MBOP | D-MPC | Diffuser, DT, TT | BC, CQL, IQL, DD, DP, IH, DBC |
| Run-time planning | ✗ | ✗ | ✓ | ✓ | ✓ | ✗ |
| Run-time novel rewards | ✗ | ✗ | ✓ | ✓ | ✓ | ✗ |
| Novel dynamics | ✓ | ✓ | ✓ | ✓ | ✗ | ✗ |
| Non-expert data | ✓ | ✓ | ✓ | ✓ | | |
| Speed at runtime | Fast | Fast | Med. | Slow | Slow | Fast |

Table 1: **A tale of three distributions**; comparing properties across offline RL methods. The methods we mention are defined as follows: MOREL etc. Kidambi et al. (2020); Yu et al. (2020; 2021); Rigter et al. (2022), Dreamer Hafner et al. (2020), DWMS (Diffusion World Models) Alonso et al. (2023), UniSim Yang et al. (2024), SynthER Lu et al. (2024), DWM (Diffusion World Model) Ding et al. (2024), PolyGRAD Rigter et al. (2024), PGD (Policy-Guided Diffusion) Jackson et al. (2024), Diffuser Janner et al. (2022), DT (decision transformer) Chen et al. (2021), TT (trajectory transformer) Janner et al. (2021), BC (behavior cloning), CQL (conservative Q learning) Kumar et al. (2020), IQL (implicit Q learning) Kostrikov et al. (2021), DD (Decision Diffuser) Ajay et al. (2023), DP (Diffuson Policy) Chi et al. (2023), IH (Imitating Humans) Pearce et al. (2023) DBC (Diffusion BC) Wang et al. (2023) .

a simple sampling-based planner that does not rely on classifier guidance. Other works using MS with a joint proposal are decision transformer (DT) (Chen et al., 2021), and trajectory transformer (Janner et al., 2021).

Similarly, the "Decision Diffuser" paper (Ajay et al., 2023) learns a trajectory distribution over states, and uses classifier-free guidance to generate trajectories that have high predicted reward; the state sequence is then converted into an action sequence using a separately trained inverse dynamics model (IDM). However, this approach does not allow for run-time specification of new reward functions.

MPC has also been applied in model-based RL, with TD-MPC (Hansen et al., 2022) and TD-MPC2 (Hansen et al., 2023) being the representative methods. D-MPC differs from the TD-MPC line of work in that D-MPC uses multi-step diffusion models for both action proposal and dynamics model, while the TD-MPC line of work uses single-step MLPs. In addition, the TD-MPC line of work focuses on online learning with environment interactions while in D-MPC we focus on learning from offline data and then use the learned models for doing MPC in the environment.

The model-based offline planning or MBOP paper (Argenson & Dulac-Arnold, 2021) was the original inspiration for our method. In contrast with the previous MPC methods, it factorizes the dynamics models and the action proposal model, which are learned separately and used at planning to optimize for novel rewards. The main difference with our work is that they use ensembles of one-step deterministic MLPs for their dynamics models and action models, whereas we use a single stochastic trajectory level diffusion model. In addition they use a somewhat complex trajectory optimization method for the action selection, whereas we use a simple sampling-based planner. Finally, we also study adapting the model to novel dynamics.

Several recent works are closely related to D-MPC. "Model-based Diffusion" (MBD) (Pan et al., 2024) also utilizes diffusion for trajectory-level search, but it assumes known environment dynamics, unlike our setting. "DyDiff" Zhao et al. (2024) employs a diffusion-based dynamics model with a structure similar to D-MPC's, conditioning on action sequences to generate state sequences. However, DyDiff's primary focus is generating synthetic on-policy data by modeling the interaction between a given single-step policy and the multi-step diffusion dynamics model, differing from our goal of planning with learned dynamics and action proposals.

D-MPC is a novel combination of MPC, factorized dynamics/action proposals, and MS diffusion modeling. This allows us to be able to adapt to novel rewards and dynamics and avoid compounding errors.

## 3 Method

We will now describe our new D-MPC method. Our approach can be seen as a multi-step diffusion extension of the model-based offline planning (MBOP) paper (Argenson & Dulac-Arnold, 2021), with a few other modifications and simplifications.

### 3.1 Model predictive control

D-MPC first learns the dynamics model $p_{s|a}$, action proposal $\pi$ and heuristic value function $J$ (see below), in an offline phase, as we discuss in Section 3.2, and then proceeds to alternate between taking an action in the environment with planning the next sequence of actions using a planner, as we discuss in Section 3.3. The overall MPC pseudocode is provided in Algorithm 1.

### 3.2 Model learning

We assume access to an offline dataset of trajectories, consisting of (state, action, reward) triples: $\mathcal{D} = \{s^1_{1:T_1}, a^1_{1:T_1}, r^2_{1:T_1}, s^2_{1:T_2}, a^2_{1:T_2}, r^m_{1:T_2}, \ldots s^M_{1:T_M}, a^M_{1:T_M}, r^M_{1:T_M}\}$. We then use this to fit a diffusion-based dynamics model $p_d(s_{t+1:t+F}|s_t, h_t, a_{t:t+F-1})$ and another diffusion-based action proposal $\rho(a_{t:t+F-1}|s_t, h_t)$. To fit these models, we minimize the denoising score matching loss in the usual way. We include a detailed review of diffusion model training in Appendix A, and refer the readers to e.g. Karras et al. (2022) for additional discussions.

We also define a function $J$ that approximates the reward-to-go given any proposed sequence of states and actions:

$$J(s_{t:t+F}, a_{t:t+F-1}) = \mathbb{E}[\sum_{k=t}^{t+F-1} \gamma^{k-t} R(s_k, a_k) + \gamma^F V(s_{t+F})] \tag{2}$$

Here $\gamma$ is the discount factor, and $V(s)$ represents the value function from state $s$ (i.e., estimate of future reward at the leaves of this search process). We also use a transformer to learn $J$ (although we can also just compute $J$ directly, if the reward function $R$ is known, and we use an admissible lower bound (such as 0) on $V$) by regressing from $(s_{t:t+F}, a_{t:t+F-1})$ to the discounted future reward in Eq. (2). We use L2 loss for the regression. We use $J$ as the objective function for optimization in MPC, and as a way to specify novel tasks. Refer to Appendix E for additional details on model architectures and hyperparameters.

---

**Algorithm 1:** Main MPC loop.

---
**1** Input: $\mathcal{D}$ = offline dataset, $N$ = num. samples, $F$ = forecast horizon, $H$ = history length
**2** $(p_d, \rho, J) = \text{train}(\mathcal{D})$
**3** $s_0 = \text{env.init}()$
**4** $h_0 = (s_0)$
**5** **for** $t = 0 : \infty$ **do**
**6**     $a_t = \text{planner.plan}(s_t, h_t, p_d, \rho, J, N, F, H)$
**7**     $(s_{t+1}, r_{t+1}) = \text{env.step}(s_t, a_t)$
**8**     $h_t = \text{append}(a_t, s_{t+1}, r_{t+1})$
**9**     $h_t = \text{suffix}(h_t, H)$

---

Note that unlike MBOP (Argenson & Dulac-Arnold, 2021) we do not need to train ensembles, since diffusion models are expressive enough to capture the richness of the respective distributions directly. Also, in contrast to Argenson & Dulac-Arnold (2021), we do not need to train a separate reward function: we estimate our value function at the beginning of the planning horizon, for a given sequence of states and actions along that horizon. In this way, our objective function $J$ already includes the estimated reward along the horizon.

### 3.3 Planning

D-MPC is compatible with any planning algorithm. When the action space is discrete, we can solve this optimization problem using Monte Carlo Tree Search, as used in the MuZero algorithm (Schrittwieser et al., 2020). Here we will only consider continuous action spaces.

We propose a simple sampling-based planner, depicted as Algorithm 2. In order to plan, given the current state $s_t$ and history $h_t$, we use our diffusion action proposal $\rho$ to sample $N$ action sequences, we predict the corresponding state sequences using $p_{s|a}$, we score these state/action sequences with the objective function $J$, we rank them to pick the best sequence, and finally we return the first action in the best sequence, and repeat the whole process. We show empirically that this outperforms more complex methods such as the Trajectory Optimization method used in the MBOP paper, which we describe in detail in the Appendix (Algorithm 5). We believe this is because the diffusion model already reasons at the trajectory level, and can natively generate a diverse set of plausible candidates without the need for additional machinery.

### 3.4 Adaptation

As with all MPC approaches, our proposed D-MPC is more computationally expensive than methods that use a reactive policy without explicit planning. However, one of the main advantages of using planning-based methods in the offline setting is that they can easily be adapted to novel reward functions, which can be different from those optimized by the behavior policy that generated the offline data. In D-MPC, we can easily incorporate novel rewards by replacing $V_n$ in Alg. 2 by $V_n = \kappa J(s_{1:F}, \mathbf{A}_{n,1:F}) + \tilde{\kappa} \tilde{J}(s_{1:F}, \mathbf{A}_{n,1:F})$, where the novel objective $\tilde{J}(s_{1:F}, \mathbf{A}_{n,1:F}) = \frac{1}{F} \sum_{t=1}^{F} f_{\text{novel}}(s_t, \mathbf{A}_{n,t})$, $f_{\text{novel}}$ is a novel reward function, and $\kappa$, $\tilde{\kappa}$ are weights of the original and novel objectives, respectively. We demonstrate this approach in Section 4.2. Of course, if the new task is very different from anything the agent has seen before, then the action proposal may be low quality, and more search may be needed.

If the dynamics of the world changes, we can use supervised fine tuning of $p_{s|a}$ on a small amount of exploratory "play" from the new distribution, and then use MPC as before. We demonstrate this in Section 4.3.

## 4 Experiments

In this section, we conduct various experiments to evaluate the effectiveness of D-MPC. Specifically we seek to answer the following questions with our experiments:

1. Does our proposed D-MPC improve performance over existing MPC approaches (which learn the model offline), and can it perform competitively with standard model-based and model-free offline RL methods?

2. Can D-MPC optimize novel rewards and quickly adapt to new environment dynamics at run time?

3. How do the different components of D-MPC contribute to its improved performance?

4. Can we distill D-MPC into a fast reactive policy for high-frequency control?

---

**Algorithm 2:** Sampling-based planner with learned multi-step diffusion action proposals

**1** Def $a = \text{Planner}(s_0, h_0, p_d, \rho, J, N, F, H)$:

**2** for $n = 1 : N$ do

**3**     $a_{n,1:F} \sim \rho(\cdot | s_0, h_0)$

**4**     $s_{1:F} \sim p_d(\cdot | s_0, h_0, a_{n,1:F})$

**5**     $V_n = J(s_{1:F}, a_{n,1:F})$

**6** $\hat{n} = \arg\max_n V_n$

**7** Return $a_{\hat{n},1}$

---

### 4.1 For fixed rewards, D-MPC is comparable to other offline RL methods

We evaluate the performance of our proposed D-MPC on various D4RL (Fu et al., 2020) tasks. Planning-based approaches are especially beneficial in cases where we do not have access to expert data. As a result, we focus our comparisons on cases where we learn with sub-optimal data. We experiment with locomotion tasks for Halfcheetah, Hopper and Walker2D for levels medium and medium-replay, Adroit tasks for pen, door and hammer with cloned data, and Franka Kitchen tasks with mixed and partial data.

Our results are summarized in Table 2. We see that our method significantly beats MBOP, and a behavior cloning (BC) baseline. It also marginally beats Diffuser, a strong model-based offline RL approach that uses classifier guidance during sampling for planning instead of MPC. In contrast to Diffuser's single diffusion model for joint state-action sequence generation, our approach decouples the dynamics model and action proposal. We note that the total compute of our two transformers is designed to be roughly equivalent to Diffuser's single model; we achieve this by using a smaller transformer for the dynamics model, reflecting its relative simplicity. We additionally compare to other popular model-free offline RL methods, like conservative Q-learning (CQL) (Kumar et al., 2020) and implicit Q-learning (IQL) (Kostrikov et al., 2021), as well as model-based RL methods like MOReL (Kidambi et al., 2020), and sequence-models like Decision Transformer (DT) (Chen et al., 2021). These methods cannot adapt to novel rewards at test time (unlike D-MPC, MBOP and Diffuser), but we include them to give a sense of the SOTA performance on this benchmark. We see that our method, despite its extra flexibility, can still match the performance of these existing, but more restrictive, methods.

| Domain | Level | MOReL | MBOP | D-MPC (ours) | Diffuser | DT | BC | CQL | IQL |
|---|---|---|---|---|---|---|---|---|---|
| halfcheetah | medium | 42.10 | 44.60 | **46.00** ($\pm$0.17) | 44.20 | 42.60 | 42.60 | 44.00 | **47.40** |
| hopper | medium | **95.40** | 48.80 | 61.24 ($\pm$2.30) | 58.50 | 67.60 | 52.90 | 58.50 | 66.30 |
| walker2d | medium | **77.80** | 41.00 | **76.21** ($\pm$2.67) | **79.70** | 74.00 | 75.30 | 72.50 | **78.30** |
| halfcheetah | medium-replay | 40.20 | 42.30 | 41.12 ($\pm$0.31) | 42.20 | 36.60 | 36.60 | **45.50** | 44.20 |
| hopper | medium-replay | 93.60 | 12.40 | **92.49** ($\pm$2.23) | **96.80** | 82.70 | 18.10 | **95.00** | **94.70** |
| walker2d | medium-replay | 49.80 | 9.70 | **78.81** ($\pm$4.19) | 61.20 | 66.60 | 26.00 | 77.20 | 73.90 |
| Locomotion Average | | **66.48** | 33.13 | **65.98** | 63.77 | 61.68 | 41.92 | **65.45** | **67.47** |

| Domain | Level | MBOP | D-MPC (ours) | DT | BC | CQL | IQL |
|---|---|---|---|---|---|---|---|
| adroit-pen | cloned | 63.20 | 89.22 ($\pm$12.57) | 71.17 ($\pm$2.70) | 99.14 ($\pm$12.27) | 14.74 ($\pm$2.31) | **114.05** ($\pm$4.78) |
| adroit-door | cloned | 0.00 | **16.36** ($\pm$2.20) | 11.18 ($\pm$0.96) | 3.40 ($\pm$0.95) | -0.08 ($\pm$0.13) | 9.02 ($\pm$1.47) |
| adroit-hammer | cloned | 4.20 | **12.27** ($\pm$3.58) | 2.74 ($\pm$0.22) | 8.90 ($\pm$4.04) | 0.32 ($\pm$0.03) | 11.63 ($\pm$1.70) |
| Adroit Average | | 22.47 | 39.28 | 28.36 | 37.15 | 4.99 | **44.90** |

| Domain | Level | D-MPC (ours) | BC | CQL | IQL |
|---|---|---|---|---|---|
| kitchen | mixed | **67.50** ($\pm$2.09) | 51.50 | 52.40 | 51.00 |
| kitchen | partial | **73.33** ($\pm$1.64) | 38.00 | 50.10 | 46.30 |
| Kitchen Average | | **70.42** | 44.75 | 51.25 | 48.65 |

Table 2: Performance comparison of D-MPC with various model-based and model-free offline RL methods across different domains. Baseline results are obtained from existing papers (Ajay et al., 2023; Tarasov et al., 2024). Performance is measured using normalized scores (Fu et al., 2020). For D-MPC, we report the mean and standard error of normalized scores over 30 episodes with different random initial environment conditions. Following (Kostrikov et al., 2021), we highlight in bold scores within 5% of the maximum per task. Baseline numbers from (Ajay et al., 2023) do not have associated standard errors. We include the standard errors for baseline numbers from (Tarasov et al., 2024) when they are present.

### 4.2 Generalization to novel rewards

In Fig. 1, we demonstrate how novel rewards can be used to generate interesting agent behaviors. We first trained the dynamics, action proposal, and value models for D-MPC on the Walker2d medium-replay dataset.

We then replaced the trained value model with a novel objective $V_n$ for scoring and ranking trajectories in our planning loop, using $f_{\text{novel}}(\boldsymbol{s}_t, \mathbf{A}_t) = 5 \exp(-(h_t - h_{\text{target}})^2 / 2\sigma^2)$, where $h_t$ is the dimension of the state $\boldsymbol{s}_t$ that corresponds to the height of the agent, $h_{\text{target}}$ is the target height, $\sigma^2 = 5 \times 10^{-4}$, $\kappa = 0$ and $\tilde{\kappa} = 1$ (so we only use the new $\tilde{J}$ and ignore the original $J$). By using this simple method, we were able to make the agent lunge forward and keep its head down ($h_{\text{target}} = 0.9$), balance itself ($h_{\text{target}} = 1.2$), and repeatedly jump ($h_{\text{target}} = 1.4$). Note that for the experiments presented in Fig. 1, we utilize a pure novel reward setting by setting $\kappa = 0$ and $\tilde{\kappa} = 1$. This means the value function used for planning, $V_n$, is solely determined by the novel reward function, $\tilde{J}$. The original reward function, $J$, derived from the pre-training data, has no influence on the agent's behavior in these cases. The resulting lunge, balance, and jump behaviors are therefore direct consequences of optimizing $\tilde{J}$ and are not present in the original dataset.

We also compared D-MPC with Diffuser on these novel reward tasks. Diffuser can also optimize these rewards, and we did not observe a significant performance difference. However, D-MPC's key advantage lies in its ability to adapt to novel dynamics, as discussed in Section 4.3.

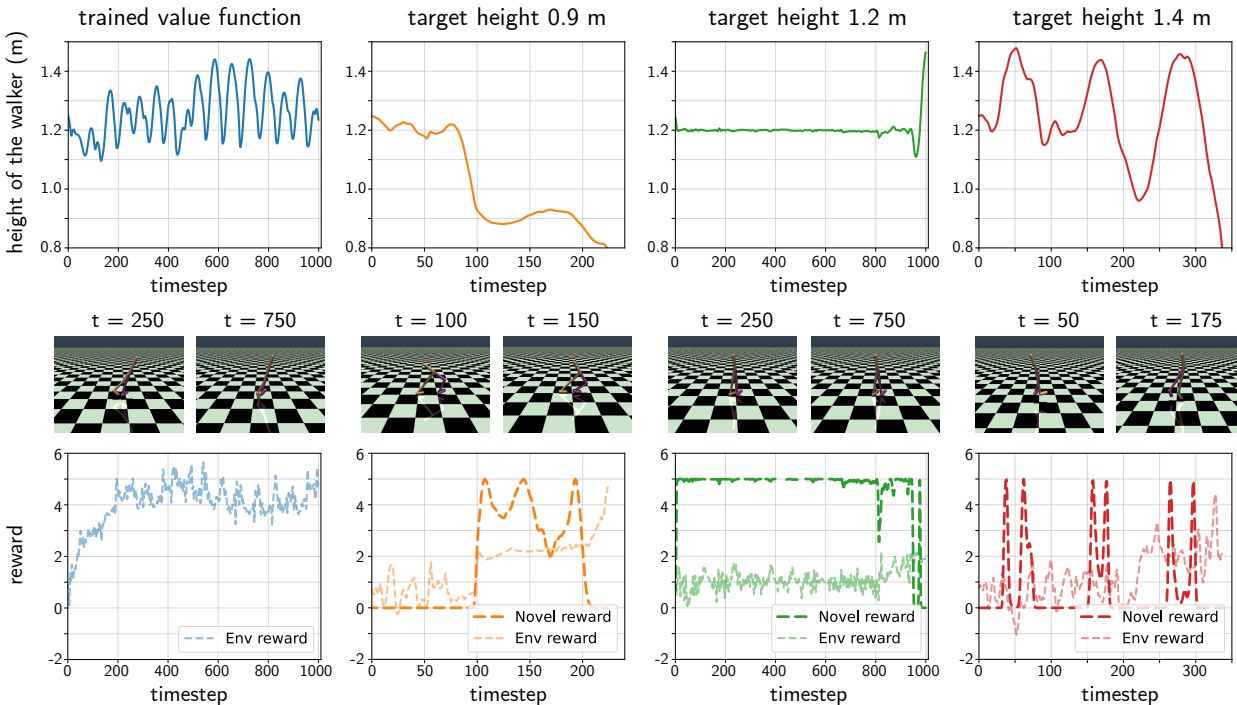

Figure 1: **Novel reward functions can generate interesting agent behaviors**. The leftmost column shows an example episode generated by D-MPC trained on the Walker2d medium-replay dataset, using the trained value function in the planner. The remaining three columns present individual examples of behaviors generated using a height-based novel objective in the planner, with each column corresponding to a different target height. The top row of each column displays the agent's height at each timestep within the episode. The middle row shows two snapshots of the agent per episode, while the bottom row graphs the novel reward (targeted by the planner) and the actual environment-provided reward received by the agent at each timestep. This figure serves as a qualitative demonstration of how novel rewards can be employed to produce interesting behaviors.

## 4.3 Adaptation to novel dynamics

In this section, we demonstrate our model's ability to adapt to novel dynamics at test time with limited experience. The need for such adaptions to novel dynamics is common in real world robotics applications where wear and tear or even imperfect calibrations can cause hardware defects and changed dynamics at test time. Because of our factorized formulation, where we separate dynamics $p_{s|a}$ from policy $\pi_a$, we can leverage a small amount of "play" data collected with the hardware defects, and use it to fine-tune our multi-step diffusion dynamics model while keeping our action proposal and trained value functions the same.

We demonstrate this on Walker2D. We train the original models on the medium dataset and simulate a hardware defect by restricting the torque executed by the actions on a foot joint (action dimension 2). On the original hardware, without the defect, trained D-MPC achieves a normalized score of 76.21 ($\pm$2.67). When executing this model on defective hardware, performance drops to only 22.74 ($\pm$1.41). Performance of our implementation of Diffuser in the same setup when deployed on defective hardware drops from 72.91 ($\pm$ 3.47) to 25.85 ($\pm$1.08).

To compensate for the changed system dynamics, we collect 100 episodes of "play" data on the defective hardware by deploying the original D-MPC trained on the medium-replay dataset. We use this small dataset to fine-tune our multi-step diffusion dynamics model, while keeping the policy proposal and value model fixed. Post-finetuning, performance improves to 30.65 ($\pm$1.89). Since diffuser jointly models state and action sequences, there is no way to independently finetune just the dynamics model. We instead fine-tune the full model with the collected "play" data. After fine-tuning, diffuser performance actually drops to 6.8 ($\pm$0.86). See Table 3a for a summary.

### 4.4 Ablation studies

In this section, we conduct a series of detailed ablation studies to illustrate how different components in D-MPC contribute to its good performance. In particular, we investigate the effect of using diffusion for action proposals, and the impact of using single-step vs. multi-step models both for the action proposals and for the dynamics models. We evaluate all variants on D4RL locomotion tasks. See Table 3b for a high-level summary of the results, and Table 4 in the appendix for detailed performances of different D-MPC variants on individual D4RL domains and levels.

#### 4.4.1 Diffusion action proposals improve performance and simplify the planning algorithm

Existing model-based offline planning methods, such as MBOP, typically use a single-step deterministic MLP policy for action proposals, an ensemble of single-step deterministic MLP models to emulate a stochastic dynamics, and rely on trajectory optimization methods for planning. This MBOP method achieves an average score of 33.13 on the locomotion tasks.

We can significantly improve on this baseline MBOP score, and simplify the algorithm, by (1) replacing their single-step MLP action proposal with a single-step diffusion proposal, and (2) replacing their TrajOpt planner with our simpler sampling-based planner. This improves performance to 52.93. Replacing their MLP dynamics with a single-step diffusion dynamics model further provides a minor improvement, to 53.32.

#### 4.4.2 Multi-step diffusion action proposals contribute to improved performance

D-MPC uses multi-step diffusion action proposals. In this section, we illustrate how this further improves performance when compared with single-step diffusion action proposals.

We start with the same single-step MLP dynamics model as in Section 4.4.1. We then replace the single-step diffusion action proposal with a multi-step diffusion action proposal that jointly samples a chunk of actions. This improves average performance from 52.93 to 57.14. We then repeated this experiment on top of the single-step diffusion dynamics, and improved performance from 53.32 to 57.81.

#### 4.4.3 Multi-step diffusion dynamics models contribute to improved performance

D-MPC uses a multi-step diffusion dynamics model. In this section we illustrate how this reduces compounding error in long-horizon dynamics prediction and contributes to improved performance.

We first measure the accuracy of long-horizon dynamics predictions of single-step diffusion, multi-step diffusion and auto-regressive transformer (ART) dynamics models (described in Appendix E.3), independent of the planning loop. We train the dynamics models on medium datasets from the respective domains, and measure the accuracy of long-horizon dynamics prediction, using state/action sequences sampled from the medium (training data), medium-replay (lower quality data) and expert (higher quality data) datasets. Concretely, we follow Schubert et al. (2023) and calculate the median root mean square deviation (RMSD) on the non-velocity

|  | Diffuser | D-MPC |
|---|---|---|
| Original | 79.60 | 76.21 (±2.67) |
| Pre-FT w/ defect | 25.85 (±1.08) | 22.74(±1.41) |
| Post-FT w/ defect | 6.8(±0.86) | 30.65(±1.89) |

(a)

| Action Proposal | Dynamics Model | | | |
|---|---|---|---|---|
|  | SS Diff | SS MLP | ART | MS Diff |
| SS Diff | 53.32 | 52.93 | - | N/A |
| MS Diff | 57.81 | 57.14 | 59.83 | **65.98** |

(b)

Table 3: (a) Performance on Walker2D before and after a simulated hardware defect, followed by fine-tuning (FT) on play data. (b) Average performances of D-MPC variants on D4RL locomotion tasks. The full D-MPC method is bottom right. MS: multi-step, SS: single step, Diff: diffusion, ART: auto-regressive transformer. Our baseline MBOP uses ensembling and MPPI trajectory optimization with SS MLP dynamics models and SS MLP action proposals, and achieves a score of 33.13.

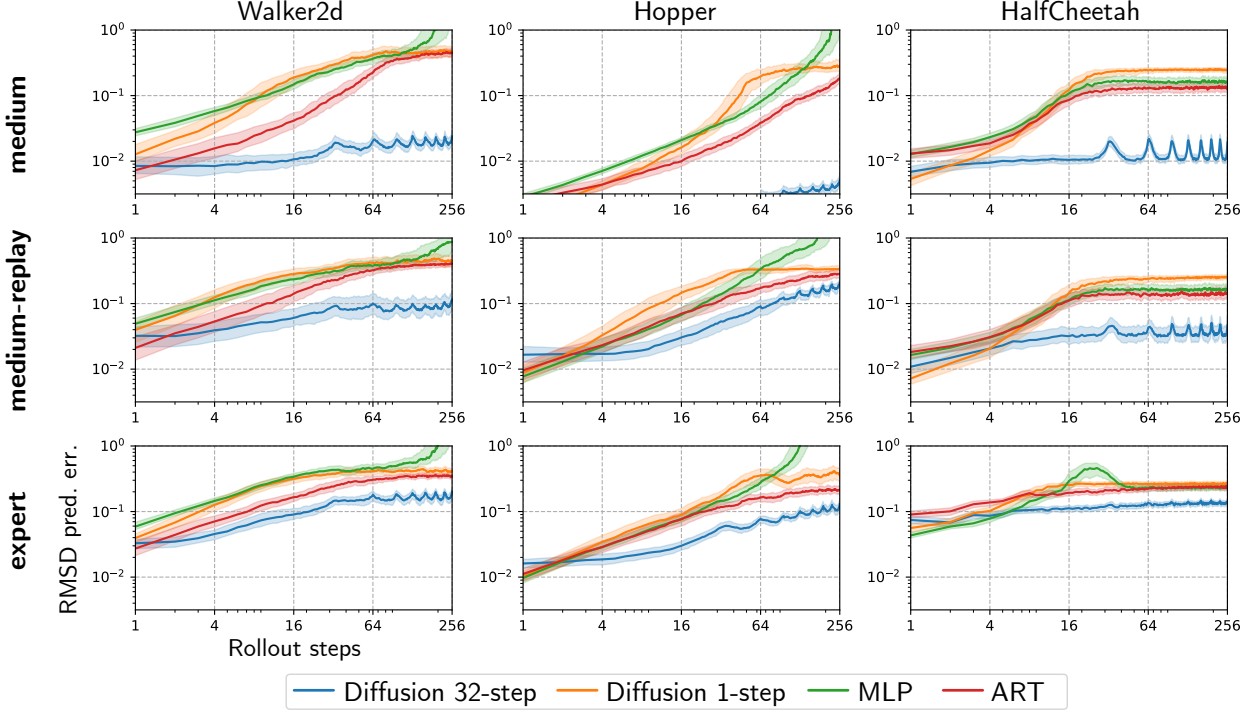

Figure 2: **Accuracy of long-horizon dynamics prediction**. We train the dynamics models on the medium dataset and evaluate on medium (training data), medium-replay (lower-quality data), and expert (higher-quality data) datasets. Prediction errors are measured by the median root mean square deviation (RMSD) on non-velocity coordinates based on 1024 sampled state action sequences of length 256. Plots show median ± 10 percentile bands. The multi-step diffusion dynamics model incurs significantly lower prediction error on training data while maintaining superior generalization abilities, outperforming other single-step and auto-regressive alternatives. The auto-regressive transformer (ART) dynamics model outperforms the single step diffusion dynamics model. The single-step MLP dynamics model exhibits compounding errors that grow rapidly for long-horizon dynamics predictions.

coordinates with increasing trajectory length. Fig. 2 summarizes the results. From the figure we can see how the multi-step diffusion dynamics model reduces compounding errors in long-horizon dynamics predictions compared to single-step and auto-regressive alternatives, while maintaining superior generalization abilities, especially for action distributions that are not too far from training distributions.

We then evaluate the quality of these dynamics models when used inside the D-MPC planning loop with a multi-step diffusion action proposal. When using the ART dynamics model, we get a score of 59.83, but when using the multi-step diffusion dynamics model, we get 65.98. We believe this difference is due to the fact that

the transformer dynamics model represents the sequence level distribution as a product of one-step (albeit non-Markovian) conditionals, i.e., $p_d(\boldsymbol{s}_{t+1:t+F}|\boldsymbol{s}_{1:t}, \boldsymbol{a}_{1:t+F-1}) = \prod_t^{t+F-1} p_d(\boldsymbol{s}_{t+1}|\boldsymbol{s}_{1:t}, \boldsymbol{a}_{1:t-1}, \boldsymbol{a}_t)$. By contrast, the diffusion dynamics model is an "a-causal" joint distribution that goes from noise to clean trajectories, rather than working left to right. We conjecture that this enables diffusion to capture global properties of a signal (e.g., predicting if the final state corresponds to the robot falling over) in a more faithful way than a causal-in-time model.

While some D4RL tasks may not present explicit, immediate penalties for short-term planning, achieving high scores even on benchmarks like Walker2D and Hopper often requires anticipating delayed consequences of actions (e.g., an excessively fast gait leading to later instability). This section demonstrates D-MPC's ability to mitigate compounding error through accurate long-horizon predictions, a crucial capability for both maximizing performance within D4RL and for tackling more complex, real-world scenarios where long-term planning is essential.

### 4.4.4 Necessity of Expressive Models for Multi-Step Prediction

To demonstrate the importance of model expressiveness for effective multi-step prediction, we replaced the multi-step diffusion models within D-MPC (for both action proposal and dynamics) with multi-step MLPs. These MLPs were two-layer networks with 4096 hidden units, using flattened multi-step states and actions as inputs. The D-MPC configuration using diffusion models achieves a normalized score of 65.98 (Table 3b). In contrast, the version using multi-step MLPs for both components only achieved a score of 50.01. This substantial performance drop underscores the necessity of employing expressive models, such as diffusion models, to accurately capture the complex distributions inherent in multi-step planning.

### 4.5 D-MPC can be distilled into a fast reactive policy for high-frequency control

Due to the use of diffusion models, D-MPC has slower runtime. In Appendix J, we include a detailed runtime comparison between D-MPC and other methods. However, if high-frequency control is important, we can distill the D-MPC planner into a fast task-specific MLP policy, similar to what is done in MoREL (Kidambi et al., 2020) or MOPO (Yu et al., 2020). Concretely, we train an MLP policy on offline datasets, using the planned actions from pretrained D-MPC as supervision. We do this for the 6 D4RL locomotion domain and level combinations we use in our ablation studies, and compare performance with both the vanilla MLP policy and D-MPC. We train all models for 1M steps, and evaluate the last checkpoint for the distilled MLP policy.

We observe that the distilled MLP policy achieves an average normalized score of 65.08 across the 6 D4RL locomotion domain and level combinations, which is only slightly worse than D-MPC's average normalized score of 65.98, and greatly outperforms the vanilla MLP policy's average normalized score of 41.92. In addition, after distillation we just have an MLP policy, and it runs at the same speed as the vanilla MLP policy.

## 5 Conclusions

We proposed Diffusion Model Predictive Control (D-MPC), which leverages diffusion models to improve MPC by learning multi-step action proposals and multi-step dynamics from offline datasets. D-MPC reduces compounding errors with its multi-step formulation, achieves competitive performance on the D4RL benchmark, and can optimize novel rewards at run time and adapt to new dynamics. Detailed ablation studies illustrate the benefits of different D-MPC components.

One disadvantage of our method (shared by all MPC methods) is the need to replan at each step, which is much slower than using a reactive policy. This is particularly problematic when using diffusion models, which are especially slow to sample from. In the future, we would like to investigate the use of recently developed speedup methods from the diffusion literature, such as distillation (see e.g., Chang et al. (2023)). Furthermore, while our ablation studies demonstrate the surprising effectiveness of our simple sampling-based planner, incorporating guided sampling techniques (as suggested in Janner et al. (2022)) could offer a path towards greater efficiency by combining the strengths of model-based and model-free approaches.

Another limitation of the current D-MPC is we only explored setups where we directly have access to the low-dimensional states, such as proprioceptive sensors on a robot. In the future, we plan to extend this work to handle pixel observations, using representation learning methods that extract abstract latent representations, which can form the input to our dynamics models, similar to existing latent-space world modeling approaches such as the Dreamer line of work, but in an MPC context, rather than a Dyna context.

Like all offline RL methods, D-MPC's performance is influenced by the distribution of behaviors in the training dataset. When offline datasets lack behaviors relevant to the target task, the generalization capabilities of any method are inherently constrained without additional data collection. While this does present a limitation for D-MPC, it is not unique to our approach but rather a fundamental challenge in offline RL. Within the scope of available data, D-MPC excels at optimizing and adapting to novel rewards and dynamics, which represents the realistic scenario for offline RL applications. Our approach's ability to effectively leverage the existing behavioral distribution is a significant strength. Future work could explore techniques to encourage broader exploration within the constraints of offline data, potentially expanding the applicability of D-MPC and similar methods to an even wider range of scenarios.

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

# A    Algorithms for model learning

In Algorithm 3 and Algorithm 4, we present the multi-step and single-step training used in our experiments.

Diffusion models are generative models that define a forward diffusion process and a reverse denoising process. The forward process gradually adds noise to the data, transforming it into a simple Gaussian distribution. The reverse process, which is learned, denoises the data step by step to recover the original data distribution.

In D-MPC, we model the action proposals and dynamics models as conditional diffusion models. Formally, let $x_0, y$ be the original data, where $x_0$ represents the data we want to model and $y$ represents the conditioning variable. Let $x_k$ be the data at step $k$ in the diffusion process. The forward process is defined as $q(x_k|x_{k-1}) = N(x_k; \sqrt{\alpha_k}x_{t-1}, (1-\alpha_k)\mathbf{I})$ where $\alpha_k$ determines the variance schedule. The reverse process aims to approximate $p_\theta(x_{k-1}|x_k) = N(x_{k-1}; \mu_\theta(x_k, k, y), \Sigma_k)$ where $N(\mu, \Sigma)$ denotes a Gaussian distribution with mean $\mu$ and covariance matrix $\Sigma$. For suitably chosen $\alpha_k$ and large enough $K$, $x_K \sim N(0, \mathbf{I})$. Starting with Gaussian noise, we iteratively denoise to generate the samples.

We train diffusion models using the standard denoising score matching loss. Concretely, we start by randomly sampling unmodified original $x_0, y$ from the dataset. For each sample, we randomly select a step $k$ in the diffusion process, and sample the random noise $\epsilon_k$ with the appropriate variance for the diffusion time step $k$. We train a noise prediction network $\epsilon_\theta$ by minimize the mean-squared-error loss $MSE(\epsilon_k, \epsilon_\theta(x_0 + \epsilon_k, k, y))$. $p_\theta(x_{k-1}|x_k, y)$ can be calculated as a function of $\epsilon_\theta(x_k, k, y)$, which allows us to draw samples from the trained conditional diffusion model.

In the D-MPC case, for learning the action proposal $\rho$, the future actions $a_{t:t+F-1}$ are the clean data $x_0$, and the current state $s_t$ and history $h_t$ are the conditioning variable $y$. For learning the multi-step dynamics model $p_d$, the future states $s_{t+1:t+f}$ are the clean data $x_0$, and the current state $s_t$, the history $h_t$ and the future actions $a_{t:t+F-1}$ are the conditioning variable $y$. In both cases, the noise prediction network $\epsilon_\theta$ is a transformer. Details of the transformer are given in Appendix E.1.

---

**Algorithm 3:** Model learning (one step models).

**1**  Def $\mathcal{M} = \text{train}(\mathcal{D}, F, H)$ :

**2**  Create dataset of tuples: $\mathcal{D}' = \{(s_t, h_t, a_t, s_{t+1}, r_t, G_t)\}$, $h_t = (s_{t-H:t-1}, a_{t-H:t-1})$, $G_t = \sum_{j=t}^{T} r_j$

**3**  Fit $p_1(s_{t+1}|s_t, h_t, a_t)$ using MLE on $\mathcal{D}'$, set $p_d = \prod_{j=t}^{t+F-1} p_1(s_{j+1}|s_j, a_j, h_j)$

**4**  Fit $\rho_1(a_t|s_t, h_t)$ using MLE on $\mathcal{D}'$, set $\rho = \prod_{j=t}^{t+F-1} \rho_1(a_j|s_j, h_j)$

**5**  Fit $R = E(r_t|s_t, h_t, a_t)$ using regression on $\mathcal{D}'$

**6**  Fit $V = E(G_t|s_t, h_t, a_t)$ using regression on $\mathcal{D}'$

---

**Algorithm 4:** Model learning (multi-step models).

**1**  Def $\mathcal{M} = \text{train}(\mathcal{D}, F, H)$ :

**2**  Create dataset of tuples:
$\mathcal{D}' = \{(s_t, h_t, a_{t:t+F-1}, s_{t+1:t+f_F}, r_t, G_t)\}$, $h_t = (s_{t-H:t-1}, a_{t-H:t-1})$, $G_t = \sum_{j=t}^{T} r_j$

**3**  Fit $p_d(s_{t+1:t+F}|s_t, h_t, a_{t:t+F-1})$ using diffusion on $\mathcal{D}'$

**4**  Fit $\rho(a_{t:t+F-1}|s_t, h_t)$ using diffusion on $\mathcal{D}'$

**5**  Fit $R = E(r_t|s_t, h_t, a_t)$ using regression on $\mathcal{D}'$

**6**  Fit $V = E(G_t|s_t, h_t, a_t)$ using regression on $\mathcal{D}'$

---

# B    The MBOP-TrajOpt Algorithm

In Algorithm 5, we include the complete MBOP-TrajOpt algorithm from Argenson & Dulac-Arnold (2021) adapted to our notations as reference.

---

**Algorithm 5:** MBOP-TrajOpt

---

**1** Def $\boldsymbol{a} = \text{MBOP-TrajOpt}(\boldsymbol{s}_0, \boldsymbol{a}^{\text{prev}}, \mathcal{M}, N, F, \kappa, \sigma^2)$:

**2** $\boldsymbol{R}_N = 0$

**3** $\mathbf{A}_{N,H} = 0$

**4** **for** $n = 1 : N$ **do**

**5** $\quad l = n \mod K$

**6** $\quad \boldsymbol{s}_1 = \boldsymbol{s}_0, \boldsymbol{a}_0 = \boldsymbol{a}_0^{\text{prev}}, R = 0$

**7** $\quad$ **for** $t = 1 : F$ **do**

**8** $\quad\quad \epsilon \sim \mathcal{N}(0, \sigma^2)$

**9** $\quad\quad \boldsymbol{a}_t = f^l_{\text{prop}}(\boldsymbol{s}_t, \boldsymbol{a}_{t-1}) + \epsilon$

**10** $\quad\quad \mathbf{A}_{n,t} = (1 - \beta)\boldsymbol{a}_t + \beta\boldsymbol{a}^{\text{prev}}_{\min(t,F-1)}$

**11** $\quad\quad \boldsymbol{s}_{t+1} = f^l_{\text{states}}(\boldsymbol{s}_t, \mathbf{A}_{n,t})$

**12** $\quad\quad R = R + \frac{1}{K}\sum_{i=1}^{K} f^i_{\text{reward}}(\boldsymbol{s}_t, \mathbf{A}_{n,t})$

**13** $\quad \boldsymbol{R}_n = R + \frac{1}{K}\sum_{i=1}^{K} f^i_{\text{value}}(\boldsymbol{s}_{F+1}, \mathbf{A}_{n,F})$

**14** $\boldsymbol{a}_t = \frac{\sum_{n=1}^{N} e^{\kappa \boldsymbol{R}_n} \mathbf{A}_{n,t+1}}{\sum_{n=1}^{N} e^{\kappa \boldsymbol{R}_n}}, \forall t \in [0, F-1]$

**15** Return $\boldsymbol{a}$

---

## C    Detailed ablation study results

In Table 4, we present detailed performances of the D-MPC variants studied in Section 4.4. See Table 3b for a a high-level summary.

| Domain Name | Level | MBOP | D-MPC | MS Diff Action Proposal | | | SS Diff Action Proposal | |
|---|---|---|---|---|---|---|---|---|
| | | | | SS Diff Dynamics | SS MLP Dynamics | ART Dynamics | SS Diff Dynamics | SS MLP Dynamics |
| halfcheetah | medium | 44.60 | 46.00 (±0.17) | 44.50 (± 0.18) | 44.78 (± 0.13) | 45.17 (± 0.15) | 46.54 (± 0.17) | 44.88 (± 0.18) |
| hopper | medium | 48.80 | 61.24 (±2.30) | 53.83 (± 2.38) | 50.66 (± 1.29) | 50.11 (± 1.77) | 46.24 (± 1.66) | 47.12 (± 2.39) |
| walker2d | medium | 41.00 | 76.21 (± 2.67) | 72.23 (± 2.49) | 77.09 (± 1.62) | 73.16 (± 2.97) | 75.59 (± 2.85) | 76.44 (± 1.79) |
| halfcheetah | medium-replay | 42.30 | 41.12 (± 0.31) | 42.85 (± 0.14) | 41.57 (± 0.16) | 42.40 (± 0.15) | 42.06 (± 0.19) | 40.37 (± 0.35) |
| hopper | medium-replay | 12.40 | 92.49 (±2.23) | 74.45 (± 4.44) | 76.38 (± 3.83) | 79.84 (± 3.93) | 70.17 (± 5.55) | 68.31 (± 5.60) |
| walker2d | medium-replay | 9.70 | 78.81 (±4.19) | 58.98 (± 4.79) | 52.33 (± 4.85) | 68.28 (± 4.43) | 39.30 (± 5.62) | 40.47 (± 5.51) |
| Average | | 33.13 | 65.98 | 57.81 | 57.14 | 59.83 | 53.32 | 52.93 |

Table 4: Detailed performances of the D-MPC variants studied in the ablation studies on different D4RL domains and levels. MS = multi-step, SS = single step, Diff = diffusion, ART = auto-regressive transformer.

## D    Normalizing state coordinates

Following Ajay et al. (2023), we normalize the states that are input to our models by using the the empirical cumulative distribution function (CDF) to remap each coordinate to lie uniformly in the range $[-1, 1]$.

Given an offline dataset of trajectories, consisting of (state, action, reward) triples

$$\mathcal{D} = \{\boldsymbol{s}^1_{1:T_1}, \boldsymbol{a}^1_{1:T_1}, r^2_{1:T_1}, \boldsymbol{s}^2_{1:T_2}, \boldsymbol{a}^2_{1:T_2}, r^m_{1:T_2}, \dots \boldsymbol{s}^M_{1:T_M}, \boldsymbol{a}^M_{1:T_M}, r^M_{1:T_M}\}$$

let $\mathcal{S}_k \equiv \left\{\bigcup_{m=1}^{M} \bigcup_{i=1}^{T_m} (s_k)^m_i\right\}$ be the accumulated corpus for the $k$-th coordinate of each state. We can define the empirical CDF for the $k$-th coordinate of the state by

$$\hat{F}_k(t) = \frac{1}{N}\sum_{i=1}^{N} \mathbf{1}_{s^i_k \leq t} \text{ for } s^{i=1\dots N}_k \in \mathcal{S}_k$$

where $\mathbf{1}_Y$ is the indicator function for event $Y$.

For the state vector $\vec{s}$ consisting of coordinates $s_k$, the relation with the normalized state coordinate is then given by $\hat{s}_k = 2\hat{F}_k(s_k) - 1$. The states output from our dynamics model are unnormalized by the inverse relation $s_k = \hat{F}_k^{-1}\left(\frac{1+\hat{s}_k}{2}\right)$.

## E  Model architectures and training details

### E.1  Diffusion models

In this paper, we train 4 kinds of diffusion models: single-step diffusion action proposals, single-step diffusion dynamics models, multi-step diffusion action proposals, and multi-step diffusion dynamics models.

We implement all 4 models as conditional diffusion models. For single-step diffusion action proposals, we use diffusion to model $p(a_t|s_t)$; for single-step diffusion dynamics models, we use diffusion to model $p(s_t+1|s_t, a_t)$; for multi-step diffusion action proposals, we use diffusion to model $p(a_{t:t+F-1}|s_t)$; and for multi-step diffusion dynamics models we use diffusion to model $p(s_{t+1:t+F}|s_t, a_{t:t+F-1})$.

Our diffusion implementation uses DDIM Song et al. (2020) with cosine schedule Nichol & Dhariwal (2021). We use transformers as our denoising network: for each conditional diffusion model, we embed the diffusion time using sinusoidal embeddings, project the time embeddings and each state and action (both for states/actions that are used as conditioning and for states/actions that are being modelled) to a shared token space with tokens of dimension 256, add Fourier positional embeddings with 16 Fourier bases to all tokens, and pass all the tokens through a number of transformer layers. We take the output tokens that correspond to the states/actions we are predicting, and project them back to the original state/action spaces.

For all our transformer layers, we use multi-headed attention with 8 heads, and 1024 total dimensions for query, key and value, and 2048 hidden dimensions for the MLP.

For all of our single-step diffusion action proposals, we use 5 diffusion timesteps and 2 transformer layers for the denoiser.

For all of our single-step diffusion dynamics models, we use 3 diffusion timesteps and 2 transformer layers for the denoiser.

For all of our multi-step diffusion action proposals, we use 32 diffusion timesteps and 5 transformer layers for the denoiser.

For all of our multi-step diffusion dynamics models, we use 10 diffusion timesteps and 5 transformer layers for the denoiser.

### E.2  One-step MLP dynamics models

We follow Argenson & Dulac-Arnold (2021) for the multi-layer perceptron (MLP) architecture of our one-step dynamics model, and train it to approximate $s_{t+1} = f(s_t, a_t)$. We use only a single MLP. Hyperparameters for the model and training are summarized in Table 5.

| Hyperparameter | Value |
|---|---|
| Number of FC layers | 2 |
| Size of FC layers | 512 |
| Non-linearity | ReLU |
| Batch size | 256 |
| Loss function | Mean square error |

Table 5: Hyperparameters for the MLP one-step dynamics model and training.

### E.3  Auto-Regressive Transformer dynamics model (ART)

We follow Chen et al. (2021) in our transformer dynamics model, leaving out the rewards tokens and the time-step embedding. Each state and action is mapped into a single token with a separate linear layer, namely $\texttt{embed}_s$ and $\texttt{embed}_a$ respectively. This results in the following tokens: $T =$

$[\text{embed}_s(\boldsymbol{s}_1), \text{embed}_a(\boldsymbol{a}_1), \text{embed}_s(\boldsymbol{s}_2), \text{embed}_a(\boldsymbol{a}_2), \ldots]$. These then normalized using `LayerNorm` Ba et al. (2016) then mapped with a causal transformer to a series of output tokens $O_1, O_2, \cdots$. The loss function is then:

$$\mathcal{L} = \sum_t \text{MSE}(\text{pred}_a(O_{2t-1}), \boldsymbol{a}_t) + \text{MSE}(\text{pred}_s(O_{2t}), \boldsymbol{s}_{t+1}) \tag{3}$$

where $\text{pred}_a$ and $\text{pred}_s$ are again linear prediction layers mapping from the output token to the state or action and `MSE` is the mean squared error.

The hyperparameters used are summarized in Table 6.

| Hyperparameter | Value |
|---|---|
| Encode dimension | 512 |
| Number of layers | 3 |
| Number of heads | 4 |
| MLP size per head | 512 |
| Attention window size | 64 |
| Position embedding | Fourier embedding with 16 basis functions. |
| Dropout | Not used. |
| LayerNorm location | Before attention block Xiong et al. (2020) |
| Non-linearity | GeLU Hendrycks & Gimpel (2016) |

Table 6: Hyperparameters used in the ART model.

### E.4 Model architectures for the objective function

Our objective function $J$ takes as input future action proposals $a_{t:t+F-1}$ and future states $s_{t:t+F}$, and regresses the discounted future reward as defined in Eq. (2). We again implement our objective function $J$ using a transformer, using the same transformer layer as in Appendix E.1. We project all states and actions to a shared token space with tokens of dimension 256, and specify an additional learnable token for the discounted future reward. We add Fourier positional embedding with 16 Fourier bases to all tokens, pass all tokens through a transformer of 10 layers, take the token that corresponds to the discounted future reward, and read out the discounted future reward prediction. We train the objective function $J$ using an L2 loss.

In our experiments for all domains except Hopper, we used a discount factor of 0.99. For Hopper we used a discount factor of 0.997. For Walker2D and Hopper, episodes can terminate early due to the agent falling down. For episodes that terminate early, we include an additional $-100$ termination penalty for the last step as reward, and calculate the discounted future reward taking into account the termination penalty.

### E.5 Training setups and hyper-parameters

For all of our model training, we use the Adam optimizer for which the learning rate warms up from 0 to $10^{-4}$ over 500 steps and then follows a cosine decay schedule from $10^{-4}$ to $10^{-5}$. We train all models for $2 \times 10^6$ steps. We use gradient clipping at norm 5, and uses EMA with a decay factor of 0.99. All of our evaluations are done using the EMA parameters for the models.

### E.6 Compute Resources

We train and evaluate all models on A100 GPUs. We use a single A100 GPU for each training run, and separate worker with a single A100 GPU for evaluation. Training for $2 \times 10^6$ steps for each variant takes about 2 days, for all variants we considered.

## F Hyper-parameters for the sampling-based planner

For all of our experiments, we use a forecast horizon $F = 32$, number of samples $N = 64$, and a history length $H = 1$. A forecast horizon $F = 32$ already works well since our trained objective function $J$ predicts discounted future rewards.

## G   Long-horizon dynamics prediction

Following Schubert et al. (2023), we measure prediction errors by the median Root Mean Square Deviation (RMSD) on the non-velocity coordinates, as depicted in Figure 2. While this metric allows us to directly analyze the effectiveness of the dynamics model, it is a somewhat crude metric of the correctness or usefulness of the prediction. For example, the following predictions would all produce errors 1.0 compared to the correct ones: treating each walker as a bundle of limbs with the same center-of-mass, predicting states that would trigger termination criteria (unreasonable joint angles), or predicting an upside down position for the HalfCheetah (a state from which it cannot recover). This metric is a more effective probe of dynamics model quality in the regime where the errors are smaller. We see in Figure 2 that the multi-step diffusion dynamics model in particular has low prediction errors even for long rollouts, indicating that it performs well in these situations.

## H   Generalization to novel rewards

For the examples in Fig. 1, we first trained the dynamics, action proposal and value components of D-MPC on the Walker2d medium-replay dataset. The leftmost column shows the agent's height and rewards attained during an example episode generated by D-MPC.

To incorporate novel rewards, we replaced the the trained value model in the planning loop with a novel objective based solely on the height of the agent. For this objective, we used $f_{\text{novel}}(\boldsymbol{s}_t, \mathbf{A}_t) = 5\exp(-(h_t - h_{\text{target}})^2/2\sigma^2)$, where $h_t$ is the dimension of the state $\boldsymbol{s}_t$ that corresponds to the height of the agent, $h_{\text{target}}$ is the desired target height, $\sigma^2 = 5 \times 10^{-4}$, $\kappa = 0$ and $\tilde{\kappa} = 1$. The scale factor of 5 in the reward function $f_{\text{novel}}$ was chosen to roughly match the maximum reward attainable in the environment.

In each episode, the agent starts at a height of 1.25 (with uniform noise in the range of $[-5 \times 10^{-4}, 5 \times 10^{-4}]$ added for stochasticity). Figure 1 demonstrates the agent's behavior for different target heights. For $h_{\text{target}} = 1.2$, which is close to the initial position, the agent maintains the desired height for an extended duration. For $h_{\text{target}} = 0.9$, the agent lowers its torso to achieve the target height, but eventually leans too far forward resulting in early episode termination. For $h_{\text{target}} = 1.4$, the agent has to jump to achieve the desired height, which can only be momentarily attained due to the environment's physics. In the example shown, the agent jumps three times before falling over, leading to early episode termination.

## I   Adaptation to novel dynamics

We used D-MPC and our implementation of the diffuser models trained on Walker2D medium dataset as our pre-trained models. To simulate defective hardware, we modified the action being executed in the environment. Specifically, we clip the action component corresponding to the torque applied on the right foot rotor to $[-0.5, 0.5]$ vs the original $[-1, 1]$.

To collect the play data on this defective hardware, we run our D-MPC model trained on medium-replay dataset, and collect data for 100 episodes. It has a total of 30170 transitions (steps) and an average normalized episode reward of 23.14 ($\pm$ 2.31). Note that the actions in this dataset correspond to the actions output by the model and not the clipped actions.

For fine-tuning, we load the pre-trained diffuser and D-MPC models and train them on this dataset with same training parameters as the original training. For D-MPC, we only train the dynamics model and freeze other components. For the diffuser, we fine-tune the full model, since it is a joint model. During online evaluation, we sampled 256 trajectories in both the models and picked the one with the best value to execute the action at each step. We report the maximum scores for both approaches.

## J   Timing measurements

To illustrate the differences in run times between different methods, we list in Table 7 the measured wall clock planning time (along with standard errors) per execution step, on a single A100 GPU, for each algorithm, in

three D4RL locomotion environments. The measurements illustrate that a simple MLP policy is the fastest, followed by an MBOP-like setup, followed by D-MPC, followed by MPC with an autoregressive transformer dynamics model. If tasks require faster control loops, D-MPC could be sped up in a few different ways such as amortizing the planning over a larger chunk of actions executed in the environment (since the planner naturally generates long-horizon plans), using accelerated diffusion sampling strategies Lu et al. (2022); Shih et al. (2024), and distilling the diffusion models Song et al. (2023); Song & Dhariwal (2023); Salimans & Ho (2022); Liu et al. (2022); Xie et al. (2024). We leave this exploration for future work.

| D4RL domain | MLP policy | MBOP | D-MPC | ART-MPC |
|---|---|---|---|---|
| Walker2d | $3.51(\pm0.06) \times 10^{-1}$ | $4.77(\pm0.03) \times 10^{0}$ | $9.37(\pm0.03) \times 10^{1}$ | $3.37(\pm0.09) \times 10^{2}$ |
| HalfCheetah | $3.73(\pm0.06) \times 10^{-1}$ | $3.93(\pm0.10) \times 10^{0}$ | $9.24(\pm0.07) \times 10^{1}$ | $2.62(\pm0.04) \times 10^{2}$ |
| Hopper | $3.49(\pm0.06) \times 10^{-1}$ | $5.07(\pm0.78) \times 10^{0}$ | $9.48(\pm0.04) \times 10^{1}$ | $3.51(\pm0.07) \times 10^{2}$ |

Table 7: Wall-clock planning time (in milliseconds) per environment step for different algorithms, as measured on a single A100 GPU.

