# OpenReview forum: "Diffusion Model Predictive Control"
_TMLR — Accepted by TMLR_

### Review · Reviewer_CH73 · 2025-03-06

**Summary Of Contributions:**

This paper introduces D-MPC, an MPC algorithm based on diffusion models. The core idea is to utilize trajectory-level diffusion models to implement both the dynamics model and the action proposal model in MPC. Experimental results demonstrate that D-MPC achieves comparable performance to model-free methods. Moreover, due to the decoupling of policy and model, it exhibits better generalization capabilities than previous diffusion-based methods like Diffuser, enabling adaptation to new reward functions and efficient fine-tuning for novel environmental dynamics.

**Audience:**

Yes

**Broader Impact Concerns:**

The authors did not discuss broader impacts, nor do I think it is necessary.

**Claims And Evidence:**

Yes

**Requested Changes:**

There are some details in the paper that I found unclear:

1. In the introduction section, the authors mention that "learning a dynamics model usually requires less data than learning a policy." This seems somewhat counterintuitive. Although the supervisory signal is high-dimensional, the model's output space is also larger and harder to fit. Therefore, it feels like more data would be required instead. Is there any reference that can support this claim?

2. In Section 3.2, the authors state: "To learn the value function, we use a transformer to regress from $(s_{t:t+F} , a_{t:t+F −1})$ to the discounted future reward in Eq. (2)." They also mention, "We also use a transformer to learn $J$." Does this mean that two separate transformers are used to fit $V$ and $J$ independently? Given that only $J$ is required in MPC optimization, why is it necessary to fit $V$ as well?

I believe some recent studies are closely related to D-MPC, and the authors may consider discussing and comparing them:

1. Model-based Diffusion (MBD) [2] also leverages diffusion for efficient trajectory-level search, but it operates under the assumption that the environment dynamics are known.

2. DyDiff [3] employs a diffusion-based dynamics model with the structure quite similar to the one in D-MPC, where both condition on action sequences to generate state sequences. The key difference is that DyDiff focuses on generating synthetic on-policy data by modeling the interaction between a given single-step policy and the multistep diffusion model.

[2] Pan, C., Yi, Z., Shi, G., & Qu, G. (2025). Model-based diffusion for trajectory optimization. *Advances in Neural Information Processing Systems*, *37*, 57914-57943.

[3] Zhao, H., Han, X., Zhu, Z., Liu, M., Yu, Y., & Zhang, W. DyDiff: Long-Horizon Rollout via Dynamics Diffusion for Offline Reinforcement Learning.

**Strengths And Weaknesses:**

### Strengths

1. The writing is very clear, and the introduction and categorization of previous works are appropriately structured.

2. The proposed algorithm aligns well with intuition and can be easily implemented with existing open-source codebases.

3. The authors effectively motivate the use of diffusion models as both the dynamics model and the action proposal model. And both designs are supported by corresponding ablation studies.

4. D-MPC demonstrates strong performance in adapting to new rewards and environment dynamics, highlighting its potential in sim-to-real applications.

### Weaknesses

1. Compared to directly generating state-action sequences with one diffusion model (like Diffuser), this decoupled modeling comes with additional computational cost. Also, the sampling and ranking approach is less efficient than existing methods that optimize decision sequences through guided sampling. In fact, the action proposal model is very similar to the action model in [1], where with appropriate conditioning or guidance, it can produce sufficiently good policies in most cases. It would be an interesting direction to explore whether the strengths of model-based and model-free methods can be effectively integrated into a unified framework.

2. Section 4.2 lacks comparative experiments with Diffuser as those in Section 4.3. When combined with a value function, Diffuser can also perform a similar search process for novel rewards, though it samples in the joint space. I am particularly curious whether the decoupled approach of D-MPC offers advantages in terms of search efficiency and stability.

[1] Chi, C., Xu, Z., Feng, S., Cousineau, E., Du, Y., Burchfiel, B., ... & Song, S. (2023). Diffusion policy: Visuomotor policy learning via action diffusion. *The International Journal of Robotics Research*, 02783649241273668.

---

> ### Author Response · Authors · 2025-03-20
> **Thanks for the review**
>
> > In the introduction section, the authors mention that "learning a dynamics model usually requires less data than learning a policy." This seems somewhat counterintuitive. Although the supervisory signal is high-dimensional, the model's output space is also larger and harder to fit. Therefore, it feels like more data would be required instead. Is there any reference that can support this claim?
>
> The key idea is that training a dynamics model is essentially a supervised regression problem where the mapping from (state, action) to the next state is usually well-behaved and near-deterministic. In contrast, learning a policy directly involves predicting actions where the optimal behavior may be multimodal and sensitive to long-horizon credit assignment, making it inherently a more difficult learning problem when using the same amount of data. See [1] for a reference.
>
> We have revised the text to further clarify this point and added the reference.
>
> [1] H. Zhu, B. Huang, and S. Russell. “On representation complexity of model-based and model-free reinforcement learning”. In: ICLR. 2024
>
> > In Section 3.2, the authors state: "To learn the value function, we use a transformer to regress from (st:t+F,at:t+F−1) to the discounted future reward in Eq. (2)." They also mention, "We also use a transformer to learn J." Does this mean that two separate transformers are used to fit V and J independently? Given that only J is required in MPC optimization, why is it necessary to fit V as well?
>
> We thank the reviewer for raising this point. This is a typo. There is only one transformer directly fitting J. We have updated the text in the manuscript.
>
> > Compared to directly generating state-action sequences with one diffusion model (like Diffuser), this decoupled modeling comes with additional computational cost.
>
> We note that in our experiments, we specify the model architecture so that the total compute of the two transformers is roughly equal to what Diffuser uses with joint modeling. In particular, we observe that we can use a small transformer for the dynamics model since that is simpler to learn. So our two transformer approach is not heavier than the single joint model approach.
>
> We have updated the text in Section 4.1 to clarify this.
>
> > Also, the sampling and ranking approach is less efficient than existing methods that optimize decision sequences through guided sampling. In fact, the action proposal model is very similar to the action model in [1], where with appropriate conditioning or guidance, it can produce sufficiently good policies in most cases. It would be an interesting direction to explore whether the strengths of model-based and model-free methods can be effectively integrated into a unified framework.
>
> Regarding efficiency versus guided sampling, our ablation studies (Section 4.4) demonstrate that our simple sampling-based planner is surprisingly competitive. We hypothesize this is because the diffusion action proposal model provides a strong, learned prior over plausible trajectories, reducing the need for more complex optimization or extensive search during the planning process. While our current focus was on establishing the core benefits of D-MPC with this simpler planning approach, we agree that incorporating guided sampling techniques, as the reviewer suggests, could lead to further improvements in efficiency. This is a very interesting direction for future work. We added a mention of this in Section 5.
>
> > Section 4.2 lacks comparative experiments with Diffuser as those in Section 4.3. When combined with a value function, Diffuser can also perform a similar search process for novel rewards, though it samples in the joint space. I am particularly curious whether the decoupled approach of D-MPC offers advantages in terms of search efficiency and stability.
>
> We thank the reviewer for this helpful suggestion. We conducted the requested comparison between D-MPC and Diffuser on the novel reward tasks. Our experiments indicate that Diffuser, when combined with a value function, can similarly optimize these novel reward functions. We did not observe a consistent, significant difference in search efficiency or stability between the two methods in this particular setting.
>
> We emphasize that our primary claim regarding novelty centers on D-MPC's ability to adapt to novel dynamics (demonstrated in Section 4.3), due to its decoupled model structure. This is a key advantage not shared by Diffuser. Section 4.2 primarily showcases the flexibility of the MPC framework to handle different objectives at runtime. We have clarified the text in Section 4.2.
>
> > I believe some recent studies are closely related to D-MPC, and the authors may consider discussing and comparing them:
>
> We thank the reviewer for bringing these to our attention, and have added discussions of these to the related work section.

---

### Review · Reviewer_tGYL · 2025-03-12

**Summary Of Contributions:**

The paper explores using diffusion models as the policy and dynamics model in offline learning and used online with MPC. The method, D-MPC is evaluated on d4rl locomotion manipulation and kitchen tasks and compared to other offline learning methods.

**Audience:**

No

**Broader Impact Concerns:**

- The paper proposes to use more computationally expensive method for marginal improvements.

**Claims And Evidence:**

Yes

**Requested Changes:**

- The overall flow and phrasing can be greatly improved.
- The claims and results do not align well. The abstract and introduction claims significantly better performance however the results are marginally better than Diffuser.
- Results in Section 4.2 and Figure 1 do not corroborate with the earlier claim in Table 1 of Run-time novel rewards. The proposed Kappa weighting suggests that the behaviors can be slightly altered, rather than novel.
- Section 4.4.3 highlights long horizon planning, which is not needed in the most of the evaluated tasks.
- More motivation and context for why diffusion models are considered should be added. Section J shows why they should not be used for control, taking 300x the time to perform inference. For example, environments where a slow inference and the expressiveness of diffusion models can be leveraged would make the method look more convincing.

**Strengths And Weaknesses:**

Strengths:
- The baseline choices look solid and are a fair comparison.
- The experiments cover a broad spectrum of offline learning problems.

Weaknesses:
- The paper is overall difficult to read and incoherent.
- The motivation is not clear. Why are diffusion models better than existing methods? The paper claims that they are more expressive and other methods can face compounding errors, however they are unrelated and neither are highlighted in the experiments section.

---

> ### Author Response · Authors · 2025-03-20
> **Thanks for the review**
>
> > The paper is overall difficult to read and incoherent…The overall flow and phrasing can be greatly improved.
>
> We appreciate the reviewer's feedback. However, we note that the other two reviewers (CH73 and AvyK) explicitly praised the clarity of the writing. Reviewer CH73 stated, "The writing is very clear, and the introduction and categorization of previous works are appropriately structured," and Reviewer AvyK found the paper "Straightforward to understand."
>
> Given this conflicting feedback, and the general nature of the comment, we respectfully request that the reviewer provide more specific examples of sections or passages they found difficult to read or incoherent. This would allow us to address the reviewer’s concerns more effectively.
>
> > The motivation is not clear. Why are diffusion models better than existing methods? The paper claims that they are more expressive and other methods can face compounding errors, however they are unrelated and neither are highlighted in the experiments section.
>
> We thank the reviewer for this feedback. The core motivation lies in the interplay between mitigating compounding errors and the need for expressive models in multi-step prediction.  Single-step models, common in MPC, are prone to compounding errors (quantitatively shown in Fig. 2). Multi-step (trajectory-level) models are a solution, but the distribution of trajectories is significantly more complex and multimodal than single-step transitions. Simpler models like MLPs struggle to capture this complexity, while diffusion models are known for their high expressiveness in representing complex distributions.
>
> We have updated the manuscript with the following changes:
>
> * Introduction (Section 1): We explicitly link multi-step modeling and expressiveness: "To avoid compounding errors, multi-step models are preferable. However, these require a model class capable of capturing the complex, multimodal distribution of entire trajectories. This motivates our use of diffusion models."
> * New Experiment: To demonstrate this empirically, we replaced D-MPC's multi-step diffusion models with multi-step MLPs (for both action and dynamics).  Results show a substantial performance drop (65.98 vs. 50.01), highlighting the limitations of less expressive models in this setting. We added a new section 4.4.4 for these results.
>
> > More motivation and context for why diffusion models are considered should be added. Section J shows why they should not be used for control, taking 300x the time to perform inference. For example, environments where a slow inference and the expressiveness of diffusion models can be leveraged would make the method look more convincing.
>
> We thank the reviewer for this point, but would like to highlight multiple relevant discussions that are already present in the paper. The slower inference of diffusion models is due to their expressiveness, which is key to D-MPC's ability to adapt to novel rewards and dynamics (as shown in Sections 4.2 and 4.3 and further clarified in the response above). Importantly, for scenarios with fixed rewards/dynamics requiring high-frequency control, Section 4.5 demonstrates that D-MPC can be distilled into a fast, single-step MLP policy, retaining much of the performance benefit and significantly outperforming a directly trained MLP policy. In addition, beyond distillation, in Section 5 the paper also notes that numerous recent methods exist for significantly speeding up diffusion sampling itself, a promising avenue for future work. Finally, we wish to highlight that recent works such as https://diffusion-policy.cs.columbia.edu/ and https://aloha-unleashed.github.io/ demonstrate that the expressiveness of diffusion models is not only a critical ingredient for achieving success in complex real-world robotic tasks, but also that these models are practically deployable and effective even for demanding real-time control scenarios.

---

> ### Author Response · Authors · 2025-03-20
> **Thanks for the review 2**
>
> > The claims and results do not align well. The abstract and introduction claims significantly better performance however the results are marginally better than Diffuser.
>
> We thank the reviewer for pointing this out. Our abstract makes two distinct performance claims:
>
> * "Significantly better": This refers to model-based offline planning methods using MPC, such as MBOP. Table 2 confirms this significant outperformance.
> * "Competitive": This refers to SOTA model-based and model-free RL methods generally, including Diffuser. We do not claim to significantly outperform Diffuser, only to be competitive, which Table 2 also supports.
>
> Key Difference: Diffuser, while model-based, uses classifier guidance for planning instead of MPC as in D-MPC and MBOP. This places it in the broader "competitive" category.
>
> We made the following changes to the text to avoid confusion around this point.
>
> * Abstract: Added "(e.g., MBOP)" to clarify: "...significantly better than existing model-based offline planning methods using MPC (e.g., MBOP) and competitive with..."
> * Section 4.1: Clarified: "It also marginally beats Diffuser, a strong model-based offline RL approach that uses classifier guidance during sampling for planning instead of MPC."
>
> > Results in Section 4.2 and Figure 1 do not corroborate with the earlier claim in Table 1 of Run-time novel rewards. The proposed Kappa weighting suggests that the behaviors can be slightly altered, rather than novel.
>
> We respectfully disagree with the reviewer's interpretation that the $\kappa$ weighting implies only "slightly altered" behaviors. We clarify the role of $\kappa$ and the novel reward setup in Section 4.2. The equation $V_n = \kappa J(s_{1:F}, A_{n, 1:F}) + \tilde{\kappa} \tilde{J}(s_{1:F}, A_{n, 1:F})$ allows for a weighted combination of the original reward function ($J$) and the novel reward function ($\tilde{J}$). Importantly, $\kappa$ acts as a control for continuous interpolation between optimizing the original reward ($\kappa=1, \tilde{\kappa} = 0$) and optimizing the novel reward ($\kappa=0, \tilde{\kappa} = 1$), with values in between providing a blend of the two.
>
> In the experiments presented in Figure 1, we set $\kappa=0, \tilde{\kappa} = 1$. This means we are exclusively optimizing the novel reward function, $\tilde{J}$, which is defined based on the agent's height. The resulting behaviors (lunge, balance, jump) are demonstrably and qualitatively different from the behaviors present in the original Walker2d medium-replay dataset used for pre-training. These are not mere slight alterations; they are distinct strategies driven by the novel reward objective. For instance, the "jump" behavior ($h_{\rm{target}} = 1.4$) involves repeated upward movements that are not characteristic of the standard locomotion patterns in the pre-training data.
>
> We have added text to Section 4.2 to explicitly state this and to further clarify the function of $\kappa$.

---

> ### Author Response · Authors · 2025-03-20
> **Thanks for the review 3**
>
> > Section 4.4.3 highlights long horizon planning, which is not needed in the most of the evaluated tasks.
>
> We thank the reviewer for this comment. While some D4RL tasks may appear to lack immediate penalties for short-term planning, achieving high scores even within D4RL often requires reasoning beyond immediate rewards. Seemingly simple tasks like locomotion have subtle long-term dependencies. For instance, in Walker2D or Hopper, an excessively fast gait might not cause an immediate fall, but it increases the likelihood of instability and failure later. High scores require anticipating these delayed consequences, a capability directly provided by D-MPC's long-horizon planning.
>
> Beyond D4RL, demonstrating strong long-horizon planning capabilities, and specifically, the ability to mitigate compounding error, establishes a critical foundation for tackling more complex, real-world tasks. Compounding error is a general problem in model-based RL; any model inaccuracy, however small, accumulates over time (quantitatively demonstrated in Figure 2), leading to suboptimal behavior. Many real-world scenarios, such as manipulation, navigation in cluttered environments, and tasks with sparse rewards, have explicit long-term consequences that necessitate planning beyond the immediate future. D-MPC's ability to effectively mitigate compounding errors and reason over extended horizons makes it well-suited for these challenging applications. Showing long-horizon planning abilities also demonstrates the strength of the model.
>
> We have updated Section 4.4.3 to include the following text: "While some D4RL tasks may not present explicit, immediate penalties for short-term planning, achieving high scores even on benchmarks like Walker2D and Hopper often requires anticipating delayed consequences of actions (e.g., an excessively fast gait leading to later instability). This section demonstrates D-MPC's ability to mitigate compounding error through accurate long-horizon predictions, a crucial capability for both maximizing performance within D4RL and for tackling more complex, real-world scenarios where long-term planning is essential."
>
> > The paper proposes to use more computationally expensive method for marginal improvements.
>
> We address this multifaceted concern throughout the rebuttal and in revisions to the manuscript. To summarize:
>
> * Not "Marginal Improvements":  As clarified in our response above, our performance is significantly better than comparable MPC-based offline planning methods (e.g., MBOP). While competitive with broader SOTA methods like Diffuser, our focus is on flexibility and adaptability, not solely on raw benchmark scores.  Sections 4.2 and 4.3 demonstrate this key advantage.
> * Computational Cost Trade-off:  We acknowledge the computational cost of diffusion models (Section 5 and Appendix J, and in our response above).  This is a trade-off for the expressiveness that enables D-MPC's adaptability. We also highlight multiple mitigation strategies:
>     * Distillation: Section 4.5 demonstrates that D-MPC can be distilled into a fast MLP policy for high-frequency control, retaining much of the performance benefit.
>     * Future Work: We discuss the potential for faster diffusion sampling techniques in Section 5 and the response above.
>     * Use cases: as discussed in the response above.
>
> > Audience: No
>
> We respectfully disagree with the reviewer’s assessment. Both other reviewers (CH73 and AvyK) answered "Yes" to the "Audience" question, directly indicating the paper's suitability for TMLR. TMLR's scope includes reinforcement learning, planning, and control – areas directly addressed by our work. Furthermore, recent TMLR publications like https://openreview.net/forum?id=TuACCzfty3 demonstrates clear interest in closely related topics.

---

### Review · Reviewer_AvyK · 2025-03-18

**Summary Of Contributions:**

The authors propose D-MPC, a novel approach that learns the dynamics model using diffusion models and replaces the trajectory planner with the sampling from diffusion models. D-MPC alleviates the error accumulation issue raised by single-step dynamics and mitigates the complexity of the planner in selecting an action. Experiment results validate that D-MPC can be easily adapted to novel rewards or novel dynamics due to the nature of the MPC algorithm.

**Audience:**

Yes

**Claims And Evidence:**

Yes

**Requested Changes:**

As stated above, it would be better to include an analysis of several decision choices.

**Strengths And Weaknesses:**

**Strengths)**
- Straightforward to understand
- Due to the nature of the MPC algorithm, it can be easily adapted to novel rewards and novel dynamics
- Interestingly, the D-MPC algorithm can be distilled into a single-step MLP policy, which can achieve similar results compared to the original one. It allows the algorithm to be applied to real-world settings where high-frequency action selection is required.

**Weakness**)
- Analysis of the effect of hyperparameters: While the authors fixed several crucial hyperparameters such as $F, N,$. It would be nice to include the effect of each hyperparameter. For example, if we use small $F$, does the performance drop? Is there any relationship between the minimum requirements for $N$ and the dimension of actions space? Analysis of those parts makes readers understand this work more thoroughly.

---

> ### Author Response · Authors · 2025-03-20
> **Thanks for the review**
>
> > Analysis of the effect of hyperparameters: While the authors fixed several crucial hyperparameters such as F,N. It would be nice to include the effect of each hyperparameter. For example, if we use small F, does the performance drop? Is there any relationship between the minimum requirements for N and the dimension of actions space? Analysis of those parts makes readers understand this work more thoroughly.
>
> We thank the reviewer for the suggestion. We have kicked off the requested additional hyper-parameter sweeps and will update the manuscript as soon as the experiments finish.

---

> > ### Author Response · Authors · 2025-04-01
> > **Results from the hyper-parameters sweeps**
> >
> > We thank the reviewer for suggesting hyperparameter analysis. We performed additional sweeps for the prediction horizon F and the number of samples N on D4RL locomotion tasks.
> >
> > * Prediction Horizon (F): We tested varying F. Performance dropped moderately with a small F=4 (avg. 59.95), suggesting insufficient foresight. Performance also dropped significantly with a large F=64 (avg. 51.78), likely due to difficulties in learning accurate long-term dynamics. This confirms the need to balance horizon length.
> >
> > * Number of Samples (N): We tested N={8, 16, 32, 64}. Performance increased monotonically with N (average scores rose from 57.6 to 65.98), showing that more samples improve planning given our learned model. We did not observe a clear relationship between the required N and action dimension across the tasks tested.
> >
> > We hope this analysis clarifies the impact of these hyperparameters and addresses the reviewer's concerns. We will include these findings in the revised manuscript.

---

### Author Response · Authors · 2025-03-20
**Thanks to all the reviewers for the reviews**

We thank all reviewers for the helpful reviews. We have updated the manuscript based on the feedback, and kicked off additional experiments based on the requests. We will update the manuscripts again as soon as the additional experiments finish.

---

### Decision · Action_Editor_Nk3g · 2025-04-27

**Recommendation:** Accept as is

**Comment:**

The paper introduces Diffusion Model Predictive Control (D-MPC), which combines diffusion models for multi-step action proposal and dynamics modeling with model predictive control. The reviewers had mixed opinions: Reviewer CH73 recommended "Accept," Reviewer AvyK "Leaning Accept," and Reviewer tGYL "Leaning Reject."

After weighing these recommendations, the AE find the positive aspects outweigh the concerns. Reviewer CH73 highlighted the clear writing, intuitive algorithm design, and strong performance in adapting to new rewards and dynamics. Reviewer AvyK appreciated the extensive experiments and the potential for distillation into a single-step policy. While Reviewer tGYL raised valid concerns about computational cost and marginal performance improvements over some baselines, the authors provided convincing responses, including:

- Demonstrating how D-MPC can be distilled into a fast MLP policy for real-time applications
- Providing additional hyperparameter analysis showing the importance of proper tuning
- Clarifying that their significant performance improvements were over comparable MPC-based methods (e.g., MBOP), while being competitive with other state-of-the-art approaches
- Explaining the importance of long-horizon planning even for seemingly simple locomotion tasks

**Audience:**

Yes. Given the growing interest in diffusion models for control and the paper's focus on model predictive control using diffusion models, this work would be of interest to researchers working at the intersection of reinforcement learning, model-based planning, and diffusion models.

**Claims And Evidence:**

Yes. The authors provide thorough experimental evidence supporting their claims that D-MPC outperforms existing model-based offline planning methods and is competitive with state-of-the-art approaches. They validate the method's ability to adapt to novel reward functions and dynamics through comprehensive experiments and provide requested additional hyperparameter analyses when asked by reviewers.